# Cryptococcal Pneumonia: An Unusual Complication in a COVID-19 Patient

**DOI:** 10.3390/diagnostics12081944

**Published:** 2022-08-12

**Authors:** Jan Štingl, Julie Hylmarová, Martina Lengerová, Jan Maláska, Jan Stašek

**Affiliations:** 1Faculty of Medicine, Masaryk University, 625 00 Brno, Czech Republic; 2Department of Pathological Anatomy, University Hospital Brno, 625 00 Brno, Czech Republic; 3Department of Internal Medicine, Hematology and Oncology, University Hospital Brno, 625 00 Brno, Czech Republic; 4Department of Anaesthesiology and Intensive Care Medicine, University Hospital Brno, 625 00 Brno, Czech Republic; 5Department of Simulation Medicine, Faculty of Medicine, Masaryk University, 625 00 Brno, Czech Republic

**Keywords:** *Cryptococcus*, COVID-19, SARS-CoV-2, fungal superinfection

## Abstract

Cryptococcal superinfection is a rare but potentially fatal complication, especially if its detection and subsequent treatment are delayed. Histopathological findings of pulmonary parenchyma from a deceased patient with these complications were acquired. Quite interestingly, only a minimal inflammatory reaction could be seen in an individual with no previously known immune suppression, indicating a disturbance of the immune system. This finding was well in concordance with the described changes in cellular immunity in COVID-19. We report the case of a 60 year old male with critical coronavirus disease 2019 (COVID-19) complicated by cryptococcal pneumonia and multiorgan failure. Both X-ray and CT scans revealed lung infiltrates corresponding with COVID-19 infection early after the onset of symptoms. Despite receiving standard treatment, the patient progressed into multiple organ failure, requiring mechanical ventilation, circulatory support, and haemodialysis. *Cryptococcus neoformans* was detected by subsequent BAL, and specific antifungal treatment was instituted. His clinical status deteriorated despite all treatment, and he died of refractory circulatory failure after 21 days from hospital admission. Histopathological findings confirmed severe diffuse alveolar damage (DAD) caused by COVID-19 and cryptococcal pneumonia. Timely diagnosis of cryptococcal superinfection may be challenging; therefore, PCR panels detecting even uncommon pathogens should be implemented while taking care of critical COVID-19 patients.

## Case Presentation

A 60 year old male smoker with a history of arterial hypertension and myocardial infarction developed typical COVID-19 symptoms in April 2021—cough, exertional dyspnoea, chest tightness, and fever. Developing prior to the widespread availability of antiCOVID-19 vaccines, he was unvaccinated. Four days after the onset of symptoms, he was admitted to the Pulmonary Department of the University Hospital Brno due to the worsening dyspnoea. He tested positive for SARS-CoV-2. The exact viral variant was not identified. The predominant variant in the Czech population was B.1.1.7 (Alpha) at that time. His initial chest X-ray revealed diffuse lung infiltrates, especially in the right upper-middle quadrant. Despite receiving standard treatment with remdesivir for five days, corticosteroids (methylprednisolone 80 mg intravenously per day), and a prophylactic dose of low-molecular-weight heparin (LMWH), his dyspnoea worsened. Hence, supplemental oxygen and antibiotics (i.e., clarithromycin 500 mg IV BID and ceftriaxone 2 g IV BID for 5 days) were administered. Due to the progressively increasing levels of D-Dimers in laboratory tests, CT angiography was performed, showing typical findings consistent with COVID-19 pneumonia of diffuse lung parenchyma involvement without any signs of pulmonary embolism (Figure 1). The patient’s oxygen was gradually increased to achieve a peripheral blood saturation of >90%. On day 11, the patient was transferred to the ICU after his saturation dropped to 50–60% despite an oxygen flow via face mask of approximately 15 litres per minute. High-flow oxygen therapy (HFOT) with awake prone positioning was commenced, and corticosteroids were switched from methylprednisolone 80 mg to dexamethasone 6 mg IV per day. Despite ten days of corticosteroid administration, the patient’s clinical condition began to deteriorate significantly, and his hypoxaemia worsened. On day 12, he was intubated and placed on invasive ventilation with aggressive parameters (positive end-expiratory pressure: 12 cm H_2_O; fraction of oxygen: 80%). Bronchoalveolar lavage (BAL) was performed with the PCR testing showing more than 2 million copies of SARS-CoV-2 per millilitre (a significant number); no fungal DNA was detected at this time. As the patient became anuric, daily dialysis treatment commenced on day 13. Ventilator-associated pneumonia (VAP) caused by *Klebsiella pneumoniae* producing extended-spectrum beta-lactamase (ESBL) was confirmed by BAL fluid cultivation, and meropenem 2 g per day in continuous infusion was started. Serum levels of cardiac markers were elevated (troponin T: 67 ng/L; NT pro Brain Natriuretic Peptide (NTproBNP): 3892 ng/L), indicating advancing myocardial injury. Norepinephrine infusion was needed to achieve adequate blood pressure. The capillary refill time was prolonged over 2 s, consistent with circulatory dysfunction. A day later, the patient developed atrial fibrillation and hemodynamic instability with doses of norepinephrine up to 0.5 µg/kg/min. On day 16, a tracheostomy was performed, while the ventilation remained fully controlled with a P/F (PaO_2_/inspiration fraction of O_2_) index below 150. Two days later, the progression of circulatory dysfunction became apparent, accompanied by an elevation of inflammatory markers. Follow-up BAL was performed, empirical vancomycin was added on day 19, and doses were adjusted respecting dialysis procedures. Abdominal ultrasound revealed no clear site of a new infection. Blood cultures were negative, and BAL showed more than 3 million copies of SARS-CoV-2 and 1200 copies of Cryptococcus neoformans per millilitre/BAL. Serum panfungal antigen ((1,3)-β-glucan D) and serum cryptococcal antigen (i.e., glucuronoxylomannan) levels were negative. Over the next four days, organ dysfunction slightly improved; therefore, the patient was slowly weaned-off sedation, and the mode of ventilation was switched to pressure support. On day 21, vancomycin was switched to linezolid 600 mg IV BID. On day 22, a follow-up BAL was performed. PCR showed borderline positivity for *Cryptococcus neoformans* (300 copies per millilitre), and serum positivity for cryptococcal antigen was detected. Combined antifungal therapy with liposomal amphotericin B (Abelcet) 500 mg IV per day and fluconazole 800 mg IV per day was commenced. Blood cultures were negative for bacteria and fungi. Cerebrospinal fluid analysis, including PCR, ruled out CNS dissemination. On day 24, the patient’s circulatory instability progressed rapidly with no response to vasopressors and inotropes, which led to a subsequent cardiac arrest followed by unsuccessful cardiopulmonary resuscitation (CPR). Multiorgan dysfunction caused by COVID-19 infection and cryptococcal pneumonia was stated as the primary cause of death from a clinician’s perspective. The most significant feature during the histopathological examination was the severe diffuse alveolar damage (DAD) (Figure 2), specifically its exudative/proliferative stage, due to the prolonged period of COVID-19 pneumonia. In addition, a considerable number of dispersed intra-alveolar microorganisms, with a thick mucus capsule, were found in the lung parenchyma (Figure 3). These microorganisms (variably sized: approximately 7–20 μm) stained with both Alcian blue (Figure 4) and Giemsa (Figure 5). There was only a very subtle inflammatory reaction in the surrounding tissue, mostly lymphocytic. A post-mortem lung smear was microbiologically tested and returned positive for *Cryptococcus neoformans*. Thus, we consider secondary lung cryptococcosis as proven. Respiratory failure as a result of DAD was the immediate cause of death of the patient.

Cryptococcosis is a significant opportunistic fungal infection. Clinical and histopathological findings in patients with normal immune status have been described as different from those in immunocompromised patients. Their findings on imaging tend to be more localised, and granulomas with monocytic infiltration can be found on histopathology examination. Patients with immunosuppression usually develop more diffuse or multifocal findings on imaging with histology showing less pronounced mononuclear inflammation and well-configured granulomas [1]. Our patient displayed very subtle histological evidence of inflammation with no granulomas upon necropsy. This can be explained mainly by the short time from diagnosis to histology, but changes in his immune status could also be a contributing factor. He tested HIV negative. Only moderate lymphopenia was present in our patient throughout his hospital stay, while normal levels of immunoglobulins were found. As we did not determine the exact levels of T- and B-lymphocytes, we can only hypothesize that the main problem could be in the cellular immune system. A certain level of immunosuppression is inherent in the early stages of COVID-19, a phenomenon consisting of changes predominantly in native immune cells and disruption of tight junctions causing epithelial injury [2]. Later stages of the disease are usually characterised by an activated immune response, a so-called cytokine storm [3]. Reduction of T-lymphocytes is well described in severe cases of COVID-19, quite interestingly corresponding with elevated levels of proinflammatory cytokines [4]. Cytokine storm has led to the implementation of several immune-modulating agents into the management of severely ill COVID-19 patients, such as corticosteroids and IL-6r (receptor for interleukin-6) inhibitors [5,6]. Our patient received treatment with methylprednisolone 80 mg intravenously for six days (equivalent to dexamethasone 15 mg), followed by dexamethasone 6 mg intravenously for four days. Low-dose steroids were repeatedly shown to be safe in ARDS patients with COVID-19 as well as in non-COVID-19 patients [7,8]. We could hypothesize that the main factor leading to immune suppression in this patient was severe COVID-19 itself. We can also only speculate whether the cryptococcal infection was acquired in the ICU based on the rapid manifestation of the disease. Still, more likely, reactivation of latent colonisation of the respiratory system developed. Regarding antifungal therapy, the combination of flucytosine and amphotericin is recommended for cases of severe cryptococcal infection [9]. As only peroral flucytosine was available at the time of diagnosis, and since the patient experienced intolerance of gastric feeding, intravenous fluconazole was chosen [10]. From a clinician’s perspective, superinfection with *Cryptococcus* spp. is rare in COVID-19 patients. Nevertheless, it must be considered, mainly due to the immune suppression inherent to COVID-19 itself. Its clinical manifestation can vary, and treatment is difficult. Therefore, establishing a timely diagnosis is crucial. This is why it is essential to routinely combine the detection of *Cryptococcus* spp. With PCR testing of bronchoalveolar lavage fluid in a deteriorating COVID-19 patient.

## Figures and Tables

**Figure 1 diagnostics-12-01944-f001:**
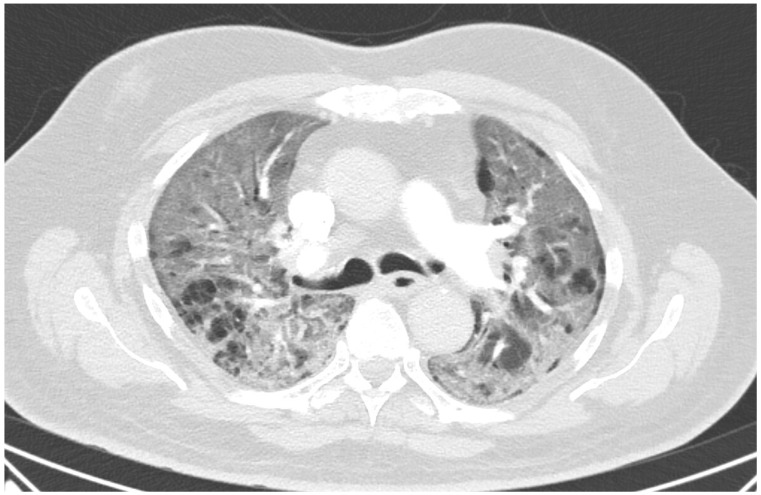
Lung CT on day 11 showing multifocal ground-glass opacities with crazy paving signs bilaterally.

**Figure 2 diagnostics-12-01944-f002:**
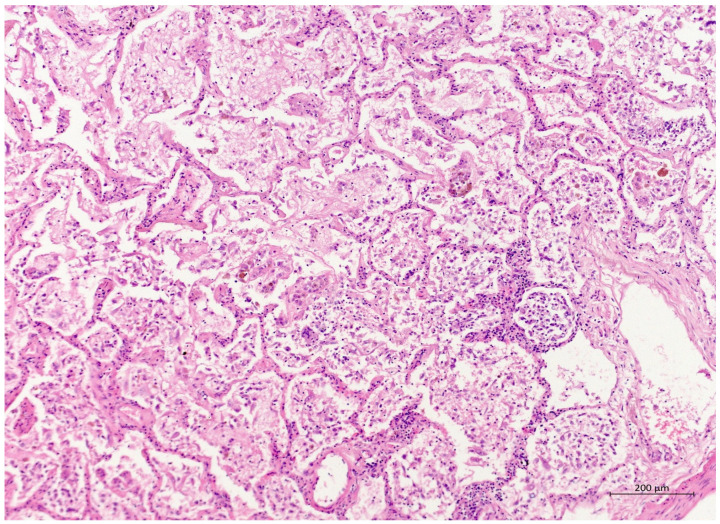
(H&E, 10×) H&E staining showing lung parenchyma with alveoli filled with exudated foamy fluid, hyaline membrane remnants, numerous desquamated cells, and mixed inflammatory infiltrate. The proliferation of myofibroblasts can be seen sporadically. The interalveolar septa were somewhat thickened.

**Figure 3 diagnostics-12-01944-f003:**
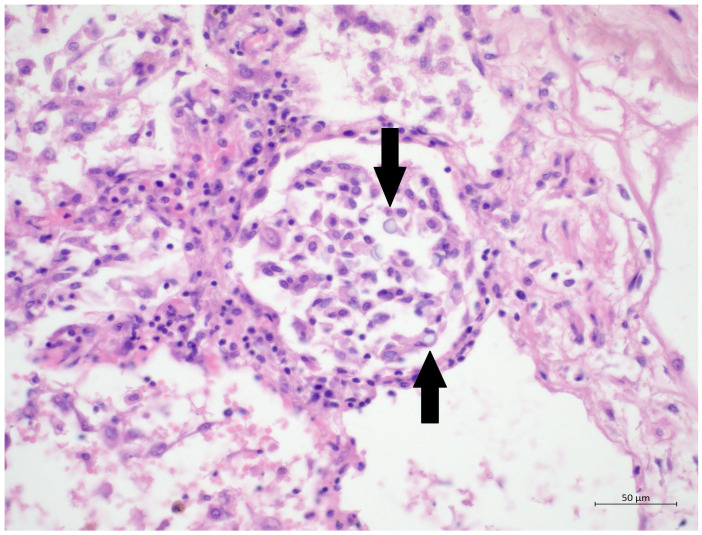
(H&E, 40×) High magnification of H&E staining, presenting alveolus with DAD-related changes; several oval to round yeasts with capsules suggestive of cryptococci are clearly visible.

**Figure 4 diagnostics-12-01944-f004:**
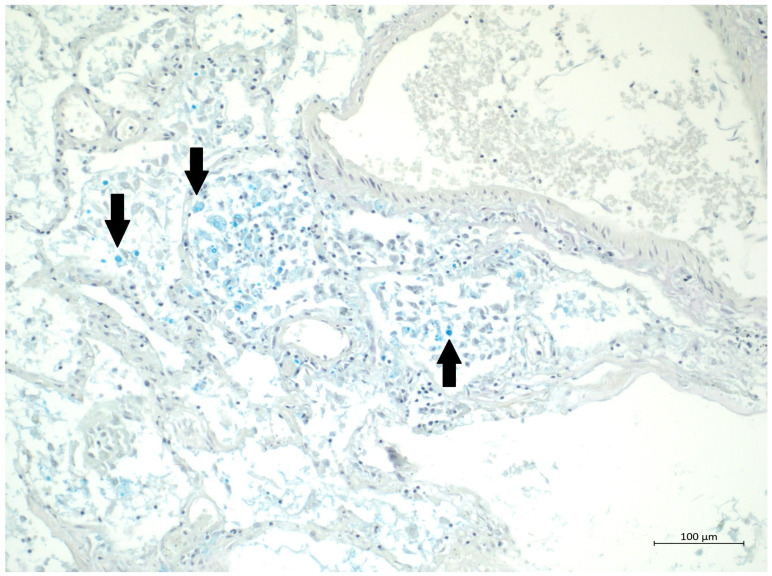
(Alcian blue, 20×) Special staining highlighting the mucous capsules.

**Figure 5 diagnostics-12-01944-f005:**
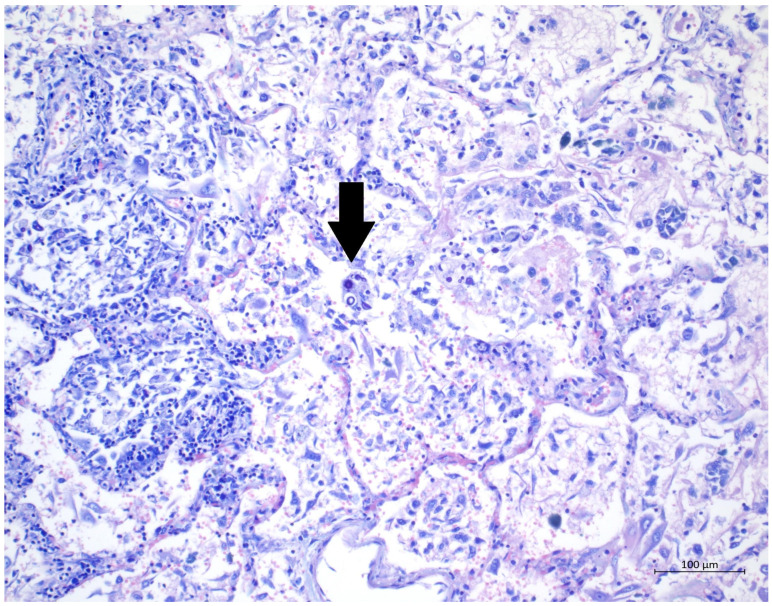
(Giemsa, 20×) Focal positivity of structures that morphologically correlated with cryptococci in H&E stain.

## Data Availability

The data presented in this article are available on request from the corresponding author. They are not publicly available due to patients’ data protection regulation.

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
