# Peer review of "Cryptococcal Pneumonia: An Unusual Complication in a COVID-19 Patient"

_diagnostics, 2022, doi:10.3390/diagnostics12081944_

Round 1

Reviewer 1 Report

Dear authors

I have only one comment regarding the Abstract section. It is recommended that the implication of this case report should be improved and represented at the end of the abstract concerning the unusual complication in a COVID-19 patient. 

Regards 

Reviewer 2 Report

This is a case describing histopathological findings related to Cryptococcus neoformans detection in   a 60-year patient with covid-19. Although report of this disease and its description is interesting, basically due to missed data or misdiagnosed disease; this report novelty is lost since there exists a previous paper exists about this same disease in a patient 60-year-old and authors propose the term CAPC (COVID-19-associated pulmonary cryptococcosis)(see Sharma et al. 2022. Indian Journal of Critical Care Medicine, 26(1)).

Authors need to determine the degree of novelty respecting to previous report.

Reviewer 3 Report

·         An interesting and well-written case report is presented

·         English language and style are fine/minor spell check required. E.g ln 16 deceased patient instead of diseased patient

·         Please comment if the patient has received any anti-cytokine drug for COVID-19 (like tocilizumab, anakinra) or Jak baricitinib.

·         Please comment why a histopathological exam was ordered. Is it standard operating procedure at your hospital? This was mainly avoided during the pandemic.

·         In Fig 1 please add a panel with the baseline CT.

·         Ln 134, 135 please add REMAP-CAP and RECOVERY references for Il-6 and SAVE-MORE for Il-1 treatment (RECOVERY Collaborative Group, Lancet 2021. REMAP-CAP Investigators, N Engl J Med 2021. Kyriazopoulou E, Nat Med 2021.)

Reviewer 4 Report

While this is an interesting case report illustrating an important clinical situation that clinicians should remain aware of, the data is presented only superficially and extensive revision is required in order to better understand the clinical case evolution reported here and the exact case timeline. Further specific comments and suggestions for improvement are given below:

-          Line 16: the term “diseased patient” should be revised.

-          Line 17-18: What basis do the authors have for their statement that low inflammation is seen in COVID-19? Phrasing should be nuanced.

-          Line 19: COVID-19 refers to the disease, therefore “COVID-19 disease” is a pleonasm.

-          Line 20: Progressive how? It would be better to specify here the COVID-19 classification, i.e., mild/moderate/severe/critical.

-          The abstract and the main text should state when this case occurred, i.e., during which COVID-19 wave, and the viral variant, either the one identified in this case, or the main variant circulating at the time in the area.

-          Also, the patient’s COVID vaccination status should be mentioned, or, if occurring prior to vaccine availability, this should also be stated.

-          Cryptococcal infection is a hallmark of HIV infection. What was this patient’s HIV status? This should be mentioned in the Abstract and the main text. It is not acceptable to miss such important information. Also, were other causes of immune suppression or immune deficit investigated?

-          Line 25: What does “standard treatment”? This statement is much too vague as during the course of this pandemic, treatment for COVID-19 has continually changed based on the newly available data from clinical trials.

-          Figure 1 and lines 45-47: I would suggest contacting a radiologist to provide help with correctly describing the image, since “diffuse changes” is far from a correct radiological description.

-          Also, is there any further follow-up imaging available?

-          Line 36: Was hypertension under treatment and therapeutically controlled? When had the myocardial infarction occurred?

-          Line 38: Progressively since when? The number of days since symptom onset should be specified.  

-          Please specify route of administration for methylprednisolone.

-          Line 41: Please specify duration of remdesivir and corticosteroids therapy. Only “a” prophylactic dose of LMWH? I.e., only one administration?

-          Line 43: Please describe oxygen flow rate per minute, and delivery method, i.e., face mask/nasal cannula, etc.

-          Line 43: Please specify dose, frequency and route of administration, and duration for each antibiotic.

-          Lines 44-45: When was CT performed during the course of disease? Please specify in the text. Is it day 8, as in the title of Fig 1? If so, is this day 8 of hospitalization? Or day 8 since symptom onset? For COVID-19 we should also always refer to the date of symptom onset.

-          Line 51: Please  specify route of administration of dexamethasone.

-          Line 53: Is “aggressive” the right term? Please specify ventilation parameters.

-          Line 53: Please define BAL at the first use in the text.

-          Line 55: What test was performed to check for “fungal DNA”?

-          Line 57, 65, 66: Please specify dose, frequency and route of administration, and duration for each antibiotic. Also, did meropenem replace the other antibiotics? Both of them of just ceftriaxone?

-          It should be clear for each change in patient status or in treatment when exactly this occurred during the course of disease. A graph should be provided to illustrate and explain this, and the text should be further clarified. For example, it is impossible to understand when BAL cultures results were received, when meropenem was added, when hemodialysis was started, etc.

-          Line 60: How was circulatory dysfunction diagnosed?

-          Line 62: Day 13 of hospitalization? Of disease?

-          Line 64: Elevation compared to when?

-          Line 67: How was Cryptococcus identified? How was the quantity calculated?

-          Line 70: “subsequent days” is too vague.

-          Lines 74-75: Please specify dose, frequency and route of administration, and duration for each drug. Furthermore, cryptococcosis is generally treated with initial therapy with amphotericin B alone or in combination with flucytosine. Fluconazole is generally used in continuation treatment. Could you justify why you chose to treat with amphotericin plus fluconazole?

-          Line 75: Negative for what? Usual bacterial cultures?

-          Linea 87-88: Microbiologically tested – which test?

-          Figure 3: “several cryptococci” is not a correct phrasing. The microbiological description should be provided instead, i.e., rounded, capsule reflection etc., and then you can state that this is suggestive for Cryptococcus.

-          Line 135: You discuss here about tocilizumab – please specify why it was not used in your reported case since there clearly would have been an indication for it.

-          Line 147: “routinely” might not be the right recommendation here. Rather, a high index of suspicion is warranted.

-          Line 166: Please define “Relative” – next-of-kin should provide consent. Furthermore, consent for images publication should also be obtained.

-          Line 164: Participation where? Since this was not a clinical trial, I do not think that this section is applicable.

-           

Round 2

Reviewer 2 Report

Author´s have attended my comment. 

Author Response

Revewers comments were attended.

Reviewer 4 Report

I thank the authors for addressing my previous comments. 

Just some final thoughts that I feel could improve the manuscript prior to publication:

1. English language and phrasing should be revised by a native speaker. 

2. Regarding my prior question, regarding flucytosine treatment, it might be worth including the author responses somewhere in the discussion paragraph. 
